# Interactive Object Grounding Using Image-Grounded Scene Graphs and Prompt Chaining

## Abstract

We introduce the task of *Interactive Object Grounding*, i.e., linking referring expressions in natural language instructions to objects in the physical environment and using clarification to handle ambiguities. Although recent foundation models can be used to perform this task in a straightforward manner, we observe that they tend to generate lengthy and sometimes confusing clarification questions. Moreover, they require many input images to fully cover complex scenes, resulting in high processing costs. Alternative approaches use a scene graph instead of images to represent the environment, but these are inhibited by relying on predefined sets of object properties and spatial relations. Instead of end-to-end VLM prompting with many images, or LLM prompting using a text-only scene graph, we propose a prompt chaining method that utilises multimodal information sampled dynamically from an *Image-Grounded Scene Graph* (IGSG), leveraging existing LLMs/VLMs to perform object grounding and clarification question generation more effectively. Evaluations based on 3D scenes from ScanNet show that the proposed method outperforms an end-to-end baseline that does not use a scene graph, at only 35% of the cost. Furthermore, it achieves substantial improvements in grounding F-score through clarification, both with our simulated user (up to 34% gain) and with human subjects (up to 23.6% gain).

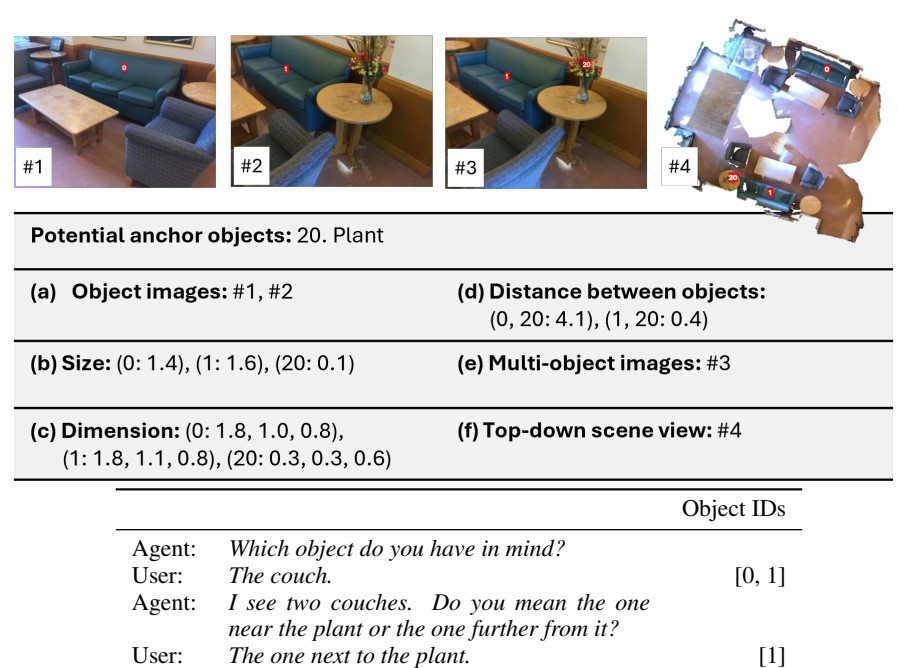

Figure 1: Example interactive object grounding dialogue with Image-grounded Scene Graph input.

## 1 INTRODUCTION

To operate effectively in a physical environment, embodied AI systems (Liu et al., 2025) require perception, reasoning, and planning capabilities. Furthermore, they should be able to learn from their interactions with the environment to become progressively more autonomous. With the arrival of the pre-trained transformer and the ensuing stream of ever more powerful foundation models, the potential capabilities of embodied AI systems seem limitless. However, the general knowledge accumulated by foundation models might not always suffice in particular settings with local specifications and requirements. In such cases, interaction with a human expert that has specialised local knowledge could bridge that gap, and help the system adapt to a new environment more rapidly.

To enable natural interaction with a human expert who is familiar with the environment but has little to no experience in robot programming, robots can be equipped with a natural language interface (Ahn et al., 2022; Park et al., 2024; Kennington et al., 2024; Quartey et al., 2024). Interpreting and following user instructions in the context of a physical environment, however, is a challenging task involving several aspects such as visual perception, natural language processing, and planning (Chai et al., 2018; Qi et al., 2019; Zhang et al., 2022; Sarch et al., 2023; Farag et al., 2025).

An important aspect of interpreting natural language instructions is *object grounding*, i.e., associating references to objects in the instruction with perceived objects in the physical environment (Achlioptas et al., 2020; Chen et al., 2020; Zhang et al., 2023; Kottur et al., 2021; Kottur & Moon, 2023). Most work in this area focuses on predicting a single object in one shot, based on a single object description. In practice, however, a description might be ambiguous, in which case the robot could engage in a clarification dialogue with the user to resolve this, i.e., perform *interactive grounding* (see Fig. 1 for an example). Although clarification in dialogue has been studied extensively (Purver et al., 2001; Schlangen, 2004; Rieser & Lemon, 2006; Stoyanchev et al., 2013; Khalid et al., 2020; Benotti & Blackburn, 2021; Aliannejadi et al., 2021; Li et al., 2024; Mazzaccara et al., 2024), in the context of embodied AI it is still an emerging topic (Gervits et al., 2021; White et al., 2021; Kottur et al., 2021; Kottur & Moon, 2023; Matsuzawa et al., 2023; Chiyah-Garcia et al., 2023).

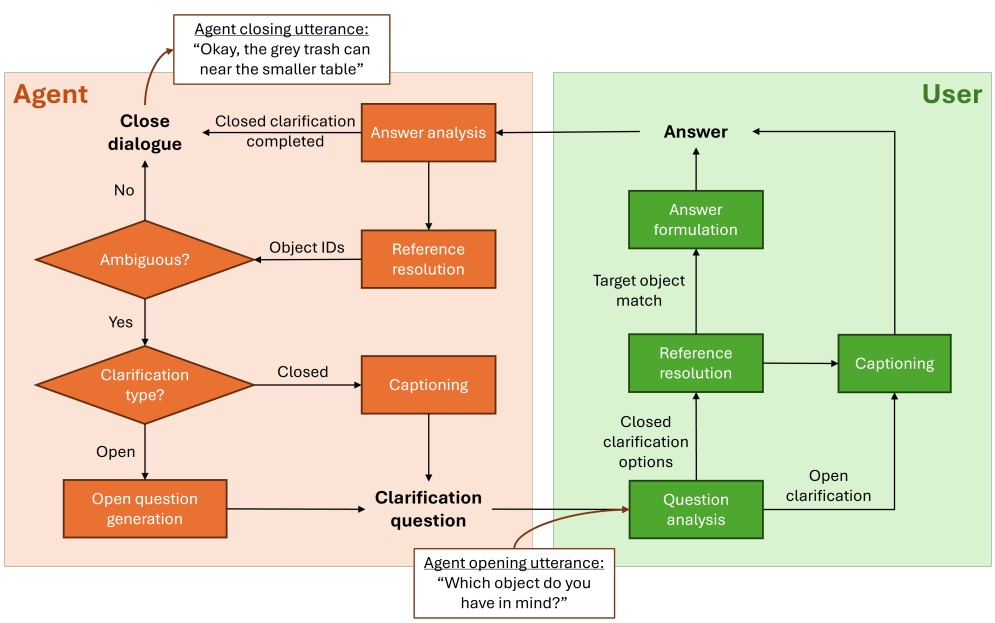

Figure 2: Interactive Grounding Agent and simulated User flowchart.

One might expect that powerful Vision Language Models (VLMs) such as GPT4o (OpenAI, 2024a) are able to perform interactive object grounding in a straightforward way, i.e., using a single prompt per dialogue turn. However, we found that in complex 3D environments, they are prone to generate lengthy, hallucinatory, or confusing responses. Moreover, their usage is costly, as they require many images of the scene to accurately interpret or generate complex object descriptions. To handle the complexity of 3D environments, alternative approaches have been developed, including fine-tuning

foundation models to process 3D input (Hong et al., 2023; Qi et al., 2025), or employing a symbolic representation of the environment in the form of a *Scene Graph* instead of images (Armeni et al., 2019; Kim et al., 2019b; Gu et al., 2024). Although these scene graph methods enable zero-shot LLM prompting, they can only handle object properties and spatial relations that can be semantically linked to the predefined properties and relations represented in the scene graph.

In this paper, we propose a novel approach to interactive grounding that addresses the above issues through a combination of prompt chaining and a new type of scene graphs. The method leverages the language understanding and generation abilities of Large Language Models (LLMs) and the reference resolution and captioning abilities of VLMs through a modular structure, implementing both the agent and a simulated user. For both reference resolution and captioning, we adopt a prompt chaining mechanism that enables dynamic infusion of multimodal information through an *Image-Grounded Scene Graph* (IGSG). Rather than fully relying on a scene graph that comprehensively represents the environment symbolically (Gu et al., 2024), this scene graph contains only basic information, such as the type, centroid, and dimensions of all objects in the scene. In particular, the graph contains an image-to-object mapping, which enables the dynamic selection of relevant images for specific VLM prompts, rather than using a fixed set of images covering the entire scene.

We have created a new benchmark based on 3D scenes from ScanNet (Dai et al., 2017), focused on identifying target objects in the presence of distractor objects of the same type, thus creating tasks that may require clarification. The proposed IGSG method has been evaluated both with our LLM/VLM-powered simulated user and with human users, demonstrating the effectiveness of the proposed clarification method. Furthermore, we show that the IGSG method outperforms an End-to-End (E2E) baseline that does not employ a scene graph, at only 35% of the cost.

In summary, we offer the following contributions:

1. An **Interactive Object Grounding** system, including (a) an *Agent* that can perform multimodal reference resolution and clarification question generation, and (b) a *Simulated User* that can answer clarification questions.

2. **Prompting methods** for dialogue analysis, reference resolution, and captioning, driven by an **Image-Grounded Scene Graph** (IGSG).

3. **Evaluations** with the simulated user and with human users; the corresponding datasets will be released upon paper acceptance.

## 2 METHOD

To leverage both the language understanding and generation abilities of LLMs (Feng et al., 2023; Chowdhery et al., 2023; Ou et al., 2024) and the captioning and reference resolution abilities of VLMs (Yu et al., 2022; Liu et al., 2023a; 2024; 2023b; OpenAI, 2024b; Chen et al., 2025; Yeshwanth & Dai, 2025; Huang et al., 2025) in zero-shot fashion, we have devised a modular structure, depicted in Fig. 2. In our experimental setup, a dialogue always starts with the agent asking the question "Which object do you have in mind?", followed by the user answering the question by describing the target object. The agent then predicts the list of candidate objects that the user might be referring to. If this results in only a single candidate object, this will be the final prediction and the agent closes the dialogue, confirming with an unambiguous object description; otherwise, the agent considers the user's object description to be ambiguous and generates a clarification question.

On both the agent and simulated user side, LLM/VLM-powered components are employed for the following tasks: 1) **Reference resolution**, 2) **Captioning**, and 3) **Dialogue analysis**. In the following, we discuss how these tasks are incorporated into the Agent and Simulated User, and then provide more details on the Image-Grounded Scene Graph and the individual components.

### 2.1 AGENT

On the agent side, **reference resolution** provides the core functionality for interactive grounding, i.e., predicting a list of object IDs given the dialogue history and information about the environment (including images of the current scene). When the agent cannot identify a unique object ID as the target, it generates an open or closed clarification question. For a *closed clarification question*,

**captioning** is used to describe the candidate objects as options for the user to choose from when answering, e.g., "Do you mean the black chair near the door or the brown chair next to the table?". An *open clarification question* summarises the ambiguity and asks the user for a more specific object description, e.g. "I see three chairs. Which one do you mean?". Currently, the agent generates a closed clarification question when there are two candidates and an open clarification question when there are more than two candidates.

## 2.2 SIMULATED USER

After the agent has generated a clarification question, the simulated user first performs **question analysis** to determine if it is open or closed. If it is open, **captioning** is used to describe the object, given the dialogue history and information about the environment. If it is closed, the alternative object descriptions (i.e., the options for the user) are extracted from the question and then **reference resolution** is performed to link them to object IDs. If one of the options matches the target object, an answer can be generated directly, e.g., "The brown chair next to the table.", in answer to the example question above; if not, the user reverts to **captioning** in order to generate a new target object description, e.g., "I mean the chair that is placed under the whiteboard.". After the user has generated an answer, the agent performs **answer analysis** to determine whether it can directly make a singleton prediction. If it can be inferred that the user has selected one of the options offered in a closed clarification question, the corresponding object ID can be used as final prediction. Otherwise, the agent will perform **reference resolution** on the answer as described above.

## 2.3 IMAGE-GROUNDED SCENE GRAPH

Where previous methods used a textual scene graph to prompt an LLM (Fang et al., 2023; Gu et al., 2024), we introduce an *Image-Grounded Scene Graph (IGSG)* to prompt a VLM. To construct a prompt for a specific VLM task, relevant information about the environment is dynamically retrieved from this graph, which may include both text and images. Our scene graph contains only basic information such as the type, location and dimensions of each object, leaving attributes like colour and spatial relations such as 'above' and 'next to' to be inferred from images in which the objects appear. More details on the proposed Image-Grounded Scene Graph are provided in Appendix B.

We use real-world scenes from ScanNet, in which the scene graphs are constructed based on the available *3D point cloud*, which itself is constructed from a set of images of the scene (Dai et al., 2017). The images fed to the VLM are annotated on-the-fly with IDs of relevant objects, depending on the prompting subtask. The IDs are positioned at the center pixel of the 2D segmentation masks that correspond to the object proposals from the point cloud, projected onto the image. The ID positioning is further improved using a high-quality 2D segmentation method (Kirillov et al., 2023).

## 2.4 REFERENCE RESOLUTION

The process of determining which object an interlocutor (the simulated user or the agent) is referring to happens in four stages. The **Expression Extraction** and **Information Specification** stages involve text-only LLM prompting, the **Information Retrieval** stage involves querying the scene graph, and the **Prediction** stage involves feeding images as well as text to a VLM. An overview of this pipeline is shown in Fig. 3, starting from the bottom left with a list of objects and a dialogue, and ending up with a prediction on the bottom right. The example describes the simulated user performing reference resolution to answer a closed clarification question asked by the agent.

In the **Expression Extraction** stage, an LLM is prompted to extract expressions from the dialogue that contain one or more references to objects, and for each expression list the relevant object IDs (see Figs. 8 and 10 for the prompts). In the example in Fig. 3 the user tries to identify the alternative object IDs for the closed clarification question asked by the agent in the second turn. To that end, the two phrases describing the alternatives are extracted from the question and combined with the IDs of all objects of the mentioned types ('trash can' and 'table'). All extracted expressions are shown on the top left in Fig. 3.

In the **Information Specification** stage, an LLM determines for each of the extracted expressions and object IDs which types of information are required to predict the referent object, selecting from the following types (see Fig. 11 for the prompt):

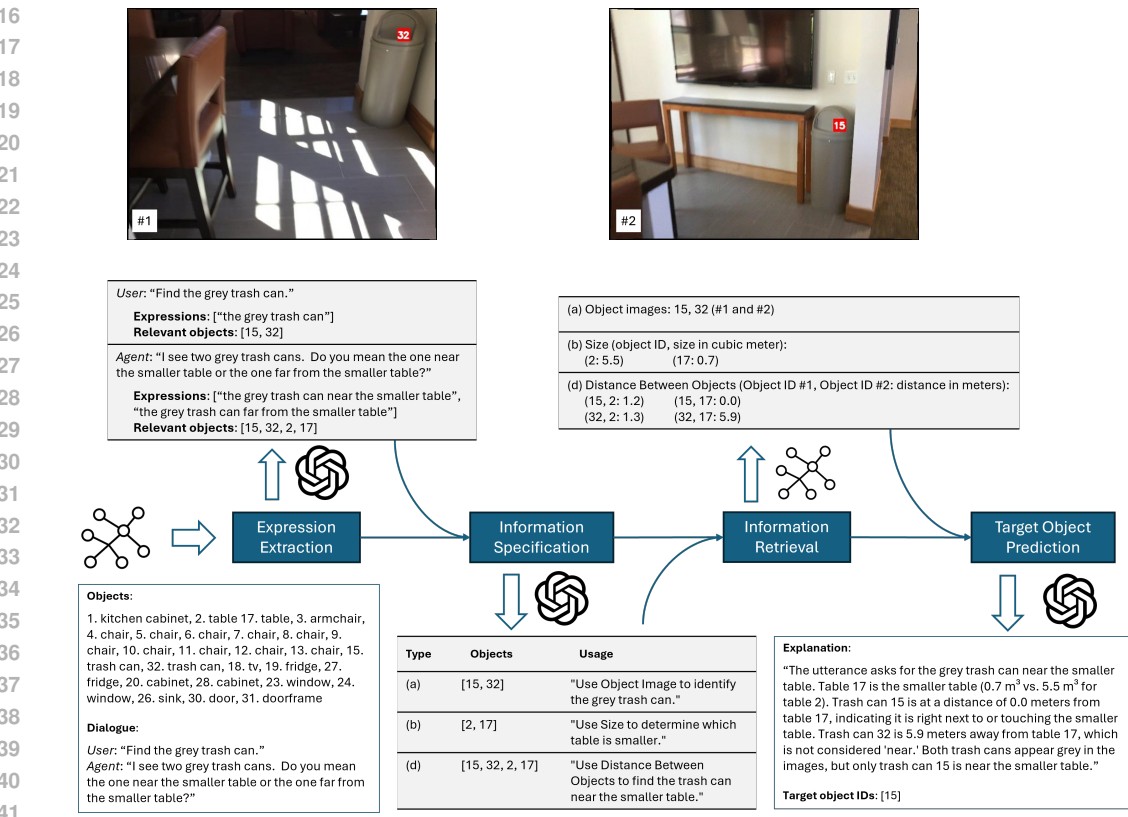

Figure 3: Reference resolution overview, indicating scene graph based and LLM/VLM based steps.

(a) **object image**: provides visual details of an object, e.g., colour, shape, etcetera,

(b) **size**: helps to distinguish between big and small objects,

(c) **dimensions**: to differentiate wide and narrow, long and short, and tall and short objects,

(d) **distance**: to indicate whether objects are near to or far from each other,

(e) **multi-object image**: provides spatial relationships, such as 'above', 'under', 'next to', 'behind', and 'in front of',

(f) **top-down scene view**: provides the overall scene layout.

The generated information specification for one of the extracted expressions is shown in the bottom centre table of Fig. 3, providing for each item the type of information (a-f), the relevant object IDs, and an explanation of how the information is to be used in the prediction stage.

In the **Information Retrieval** stage, the actual information according to the generated specification is retrieved from the scene graph and passed on to the **Target Object Prediction** stage. In this final stage, a VLM is prompted (see Fig. 12) to generate a target object prediction, in the form of a list of candidate object IDs, along with an explanation, as exemplified on the bottom right in Fig. 3.

## 2.5 CAPTIONING

Generating object descriptions for the purpose of simulating user answers or constructing closed clarification questions is done in two stages: **Information Sampling** from the scene graph and **Object Description Generation** based on the sampled information. An overview is given in Fig. 4.

During **Information Sampling**, we heuristically extract information from the scene graph that might be useful in describing a given target object. In addition to the target and its distractors (other objects of the same type), potential anchor objects are included, restricted to object types with only one instance to avoid confusion. The table on the bottom left of Fig. 4 shows the sampled scene

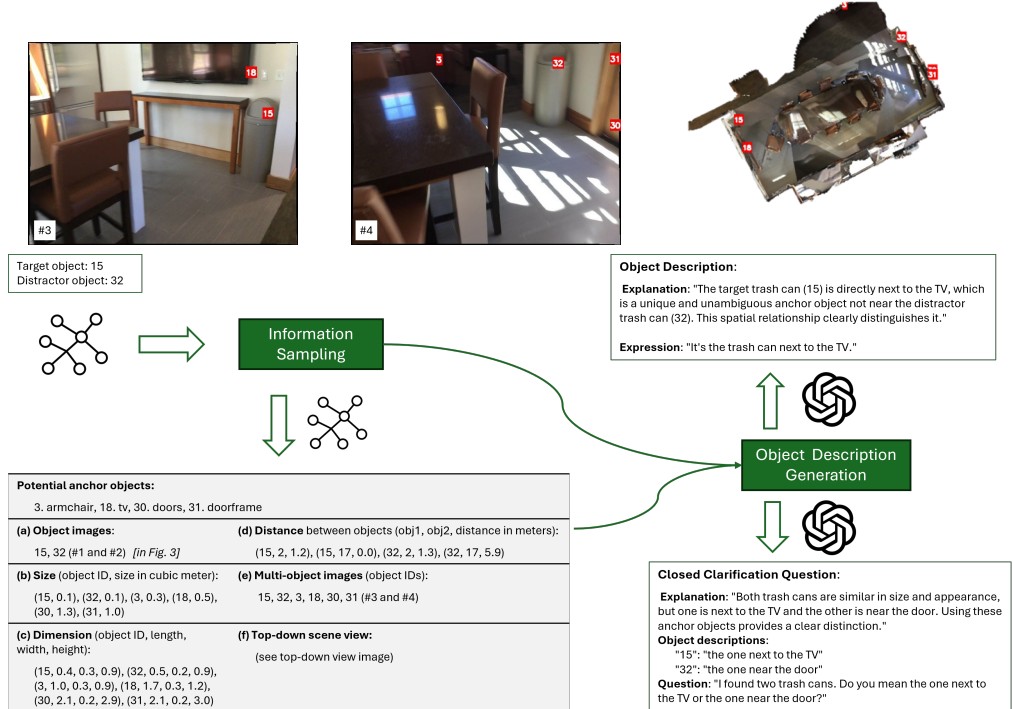

Figure 4: Captioning overview, indicating scene graph based and LLM/VLM based steps.

graph information for an example with a target and distractor of type 'trash can', adding four objects as potential anchors. For the selected objects, the information of all the types (a-f) is extracted, selecting multi-object images so that anchors appear together with the target/distractor objects.

In the **Object Description Generation** stage, a VLM is prompted (see Fig. 13, 14 and 15) with the sampled scene graph information, including the selected images, to generate an object description (see the example on the top right in Fig. 4) or a closed clarification question (see the example output on the bottom right in Fig. 4) describing a limited number of alternative objects.

### 2.6 DIALOGUE ANALYSIS COMPONENTS

The three remaining components involve text-only prompting of an LLM with the dialogue history to perform a task. In **Question analysis**, the simulated user prompts the LLM (see Fig. 9) to decide whether the agent asked an open or closed clarification question. In **Answer formulation**, the simulated user prompts the LLM (see Fig. 17) to generate an answer to a closed clarification question, based on reference resolution results on the question. In **Answer analysis**, the agent prompts the LLM (see Fig. 7) to decide if the target object can be inferred directly from the user answer, given the dialogue history.

## 3 EXPERIMENTS

To evaluate the proposed interactive grounding method, we have created a dataset based on scenes from ScanNet (Dai et al., 2017) and have collected dialogues, using the simulated user described above as well as human users. Interactive grounding performance is measured in terms of precision, recall, and F-score of the predicted object IDs against the ground-truth target object ID. Aiming for a high recall on ambiguous object descriptions, the clarification dialogue should help narrow down the list of candidate objects, thus improving precision. We compare performance levels of the IGSG system before and after clarification in various conditions, and also compare the IGSG system with a baseline that does not employ a scene graph and performs reference resolution and clarification

end-to-end, i.e., using a single prompt. For all components, the 'gpt-4.1' model is used [1], with the *temperature* and *top_p* hyperparameters both set to 0. All prompts can be found in Appendix C.

## 3.1 DATA

We selected 65 scenes from ScanNet and generated the corresponding scene graphs, using the available 3D point cloud information. In each scene, we identified the object types that had at least two, and up to four instances, and randomly selected one instance of each of those types as a target object. Hence, each target object had at least one and up to three distractors. This process resulted in a dataset of 263 instances, with an average of 1.33 distractors per instance.

## 3.2 END-TO-END BASELINE

For comparison, we created a baseline system which uses the same VLM, but in a much more straightforward way, using a single prompt (see Fig. 6) to perform both reference resolution and clarification question generation. Furthermore, this end-to-end system does not employ a scene graph, but is used out-of-the-box by feeding it a set of images in which every object appears at least once, in addition to a top-down view image for the VLM to understand the overall scene.

## 3.3 EVALUATION WITH THE SIMULATED USER

We first used the simulated user to automatically evaluate the proposed Image-Grounded Scene-Graph (IGSG) method and the End-to-End (E2E) baseline method on the full dataset. Three different conditions were created, determined by the behaviour of the simulated user in the first turn:

1. **ambTp**: ambiguous answer by only referring to the target object type, e.g. "it is a chair",

2. **ambLm**: LLM generated answer, intended to be ambiguous, e.g. "the grey trash can",

3. **unamb**: LLM generated answer, intended to be unambiguous, e.g. "the grey trash can near the smaller table".

The results in Table 1 show that the clarification method greatly improves performance in all conditions. We can also see that forced ambiguity in the first turn (ambTp) results in the lowest precision and highest recall before clarification, improving to the highest precision and highest recall after clarification. Without forcing or encouraging ambiguity in the first user turn (unamb), precision before clarification is much higher, confirming that the generated object descriptions are more specific. Using the VLM to generate ambiguous user answers in the first turn (ambLm) results in low precision as expected, but also in lower recall. This may be due to these answers sometimes being too vague, resulting in the system failing to predict any candidate objects. This low recall also limits the narrowing down potential of clarification, as the lower gain scores demonstrate.

The results also show that the IGSG agent clearly outperforms the E2E agent in terms of clarification, given the similar scores before, but much higher scores after clarification. Moreover, the IGSG agent uses much less computation time (25 vs 49 seconds per dialogue) and has much lower VLM usage costs ($0.018 vs $0.052 per dialogue).

In the AUTOEVAL:SUBSET part of Table 1, we report results on the same data subset that was used for the human evaluation. These results show a similar pattern to the full dataset results, except for the markedly lower precision in the ambiguity conditions. This can be explained by the fact that the average number of distractors turned out to be higher in this subset (1.66 vs 1.33), see Table 2.

Table 3 shows (partial) dialogues generated with the simulated user in interaction with the E2E system (left) and with the proposed IGSG system (right) for an example from the dataset. Where the E2E system feeds eight images to the VLM to generate a clarification question, our system only uses four (see also Fig. 1). Furthermore, the question generated by the E2E system is confusing due to its misguided reference to the top-down view ('upper right area', 'lower left area'). Instead, the IGSG system uses distance to an anchor for contrasting the candidate objects in generating the question.

---

[1] https://openai.com/index/gpt-4-1/

Table 1: Results in terms of target object prediction **P**recision, **R**ecall, and **F**-score, comparing the IGSG and E2E agents, and user ambiguity conditions ambTp (object **Typ**e), ambLm (**LLM** based), **unamb**iguous, and Forced/Free in the human evaluation.

| Dataset | Condition | Before clarification | | | After clarification | | | Gain |
|---|---|---|---|---|---|---|---|---|
| | | **P** | **R** | **F** | **P** | **R** | **F** | **F** |
| AUTOEVAL FULL DATASET | ambTp-E2E | 42.1 | 100 | 59.3 | 88.3 | 89.0 | 88.6 | +29.3 |
| | ambTp-IGSG | 41.5 | 99.6 | 58.6 | 92.8 | 92.8 | **92.8** | **+34.2** |
| | ambLm-E2E | 43.5 | 82.1 | 56.9 | 75.9 | 75.3 | 75.6 | +18.7 |
| | ambLm-IGSG | 44.2 | 81.0 | 57.2 | 75.5 | 77.2 | 76.3 | +19.1 |
| | unamb-IGSG | 84.9 | 91.6 | **88.1** | 91.5 | 90.1 | 90.8 | +2.7 |
| AUTOEVAL SUBSET | ambTp-E2E | 37.9 | 100 | 55.0 | 89.7 | 90.9 | 90.3 | +35.3 |
| | ambTp-IGSG | 36.5 | 100 | 53.5 | 90.9 | 90.9 | 90.9 | **+37.4** |
| | ambLm-E2E | 38.5 | 77.9 | 51.5 | 73.7 | 72.7 | 73.2 | +21.7 |
| | ambLm-IGSG | 41.7 | 84.4 | 55.8 | 74.4 | 79.2 | 76.7 | +20.9 |
| | unamb-IGSG | 84.7 | 93.5 | **88.9** | 92.1 | 90.9 | **91.5** | +2.6 |
| HUMANEVAL SUBSET | Forced-IGSG | 36.7 | 97.5 | 53.3 | 77.9 | 75.9 | **76.9** | **+23.6** |
| | Free-IGSG | 47.7 | 79.7 | **59.7** | 73.4 | 73.4 | 73.4 | +13.7 |

Table 2: Evaluation dataset statistics.

| Dataset | #Scenes | #Tasks | #Distractors per task |
|---|---|---|---|
| Full dataset | 65 | 263 | 1.33 |
| Human evaluation subset | 45 | 79 | 1.66 |

## 3.4 EVALUATION WITH HUMAN USERS

To verify the evaluation results from the simulated user, we carried out a small scale human user experiment, recruiting 8 subjects from within our organisation. During the experiment, for each task, the user was presented with a series of images from a ScanNet scene, annotated with object IDs. The user was also shown a list of object IDs with their object types, which they could use to refer to objects in the scene. In addition, the user could get an overview of the scene as a whole by opening a visualisation of the point cloud[2]. Finally, the user was given a target object ID and asked to answer the agent's questions accordingly. An example scenario for the human user experimental setup can be found in Appendix A.

We randomly selected 79 instances from the full test set and collected dialogues for them in two different conditions:

- **Forced ambiguity**: in their first answer, the user should only refer to the target object type, creating an ambiguous expression (roughly corresponding to the ambTp condition).

- **Unrestricted**: the user simply conveys the identity of the target object to the agent, using one or more turns of natural conversation (roughly corresponding to a mixture of the ambLm and unamb conditions).

Every subject carried out ten tasks (corresponding to ten different instances) in both conditions, each producing a dialogue for one of the 79 instances. After collecting all dialogues, including the agent's target object predictions, precision, recall, and F-scores were calculated.

The human user evaluation results in Table 1 show that the best performance is achieved in the forced ambiguity condition, emphasising the importance and effectiveness of the clarification methodology. The users seemed to benefit from starting with an ambiguous object description in the first turn and then answering clarification questions generated by the agent, rather than immediately trying to uniquely describe the target object. The results also suggest that compared to the simulated user,

---

[2]https://kaldir.vc.in.tum.de/scannet_browse/scans/scannet/querier

Table 3: Comparing clarification questions from the E2E baseline and the proposed IGSG system.

| E2E system | IGSG system |
|---|---|
| **Agent**: Which object did you have in mind? | |
| **User**: The couch. | |
| **Agent**: Is the couch you are referring to the one in the upper right area of the room, or the one in the lower left area of the room? | **Agent**: I see two couches. Do you mean the one near the plant or the one further from it? |
| **Explanation**: *The couches can be distinguished by referencing their positions in the room.* | **Explanation**: *Both couches are similar in size and appearance, but one (ID 1) is near a plant (ID 20) (0.4m) and the other (ID 0) is not (4.1m). The couch near the plant is clearly distinguishable.* |

the human users struggled to grasp the complex 3D scenes and produce sufficiently accurate object descriptions to help the agent identify the target object.

## 4 RELATED WORK

There are various benchmarks for clarification in multimodal settings: for the IGLU competition (Kiseleva et al., 2021; 2022) the MineCraft corpus (Narayan-Chen et al., 2019) was extended with clarification questions, Madureira & Schlangen (2023) have provided annotations of instruction clarification requests (iCRs) in the CoDraw dataset (Kim et al., 2019a), and in (Kottur & Moon, 2023), multimodal reference resolution tasks were defined for both ambiguous and unambiguous object mentions, but no specific task for clarification was included. Similarly, Haber et al. (2019) offer a dataset of visually grounded dialogues and a dialogue-aware reference resolution baseline, without focusing specifically on clarification for handling ambiguities. Our benchmark uses complex 3D scenes from ScanNet, focusing on clarification question and answer generation to handle ambiguous object descriptions.

Other approaches to multimodal clarification question generation include White et al. (2021) who trained a modular system evaluated on the 20 questions game, which is limited to yes/no questions only, and Matsuzawa et al. (2023), who trained a model for generating questions, but did not evaluate it in an interactive setting with a user answering questions in a clarification dialogue.

There are many papers in the area of 3D scene understanding, reporting results on various tasks, including reference resolution, captioning and question answering. SeeGround (Li et al., 2025) proposed a reference resolution approach that also leverages VLMs, but using query-aligned *rendered* images to prompt them, rather than *raw* images selected through an Image-grounded Scene Graph as we propose. Numerous methods for 3D object captioning have also been proposed (Luo et al., 2023; Huang et al., 2024b;a), but none of them have been used for clarification question generation. Although some works have proposed unified frameworks to perform a range of 3D comprehension tasks, what we have proposed here is an approach that combines some of these tasks into a single interactive system, providing both an agent and a simulated user.

## 5 CONCLUSION

In this paper, we have introduced the task of interactive object grounding, combining reference resolution and clarification with an expert user to efficiently identify target objects in 3D environments. Our proposed method for this task is characterised by an image-grounded scene graph providing dynamically sampled multimodal information, leveraging existing LLM/VLM capabilities in zero-shot fashion through prompt chaining. In evaluations on a custom dataset based on 3D scenes from ScanNet, we have demonstrated that the proposed method effectively improves grounding performance through clarification with both simulated and human users, and also outperforms an end-to-end baseline method (using the same VLM), and at significantly lower cost.

## 6 ETHICS STATEMENT

Part of our evaluation experiments involved a small group of human volunteer subjects, recruited from within our organisation; see Section 3.4. Their data consists solely of typed user utterances, which will be publicly released in anonymised form with their permission.

## 7 REPRODUCIBILITY STATEMENT

In this paper, we have provided detailed descriptions of our proposed interactive grounding agent and simulated user, as well as the end-to-end baseline system (Section 2). For the VLM-based steps, we have specified which models were used in the evaluation, and confirmed hyperparameter settings that make the VLM responses as consistent as possible (Section 3). The full prompts of all LLM/VLM-based steps have been included in Appendix C. Finally, we will release the full evaluation dataset upon acceptance of the paper.

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

## A  HUMAN USER EVALUATION EXAMPLE SCENARIO

Figure 5 shows an example scenario with multiple objects of type 'table', only one of which is the target object, presented to the user. In the dialogue, the user first describes the target object by referring to its type only, which results in the agent predicting all three objects of type 'table' as candidates. After the agent asks an open clarification question, the user uses a spatial relationship with an anchor object in their answer, resulting in the agent correctly identifying the object with ID 8 as the target.

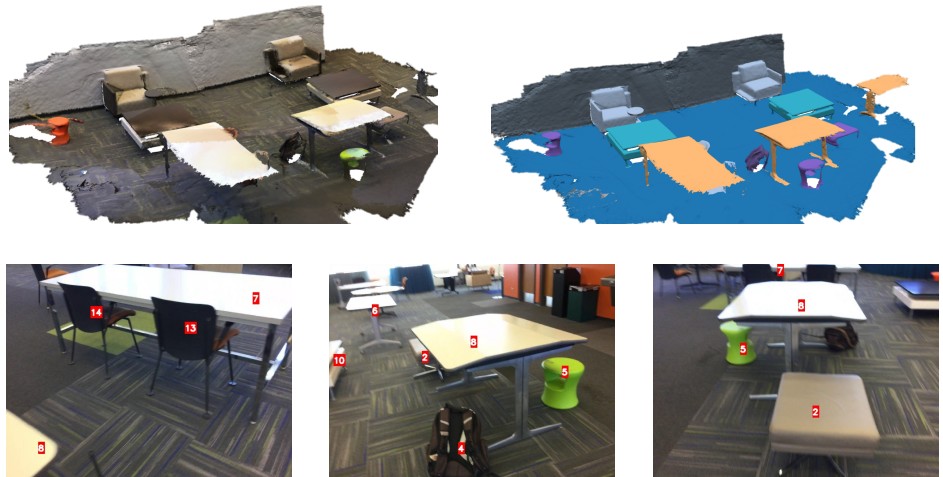

**Objects**: 0. chair, 1. chair, 2. seat, 3. floor, 4. backpack, 5. seat, *6. table*, *7. table*, 8. table, 9. coffee table, 10. coffee table, 11. wall, 12. wall, 13. chair, 14. chair, 15. seat

| Agent utterance | User utterance | Agent prediction |
|---|---|---|
| "Which object did you have in mind?" | "The table" | [6, 7, 8] |
| "I see three tables, which one do you mean?" | "the one next to a stool" | [8] |

Figure 5: Example scenario, showing two views of the point cloud, three images from the real world scene, annotated with object IDs, the list of objects known to the agent, the target object underlined and the distractors italicised, and a dialogue in which the agent interactively identifies the target object (8) known to the user.

## B  CONTENTS OF IMAGE-GROUNDED SCENE GRAPH

We propose a scene graph with nodes representing objects and edges representing relations between objects, as usual. However, we only store basic information about each object and rely on the raw images of the scene to cover other object properties and relations. To facilitate this, the scene graph

is grounded in the images via an image-to-object mapping, a JSON structure of 2D object centroids in each image, used to generate object ID annotations for VLM prompting.

Object properties:

- **ID**: Unique identifier for the object.
- **label**: Object type.
- **centroid**: 3D coordinates of the object's centroid.
- **dimension**: Length, width, and height of the 3D bounding box (L/W/H).
- **size**: Overall size of the object.
- **image_indices**: List of image indices where the object appears.
- **occupancies**: List of the object's 2D-to-3D point projection ratios in each image (number of 3D points projected onto a 2D image ÷ total number of 3D points), used to select object or multi-object images for VLM reasoning.
- **areas**: List of the object's 2D areas in each image, used to select object or multi-object images for VLM reasoning.
- **pointcloud**: 3D point cloud of the object, used to calculate distance between objects and produce top-down scene views.

## C PROMPTS

### C.1 END-TO-END BASELINE AGENT

The End-to-end baseline agent uses a single prompt, shown in Fig. 6

```
You are a helpful assistant specialized in 3D visual grounding.

I will provide a dialogue between an user and an agent, in which the user attempts to
locate an object within a scene with the help of the agent. Your task is to identify
the target object's ID from a list of objects, based on the user's utterances, the
images of the scene, and a top-down view of the scene.

If more than one candidate object likely satisfies the conditions described by the
user, include ALL such candidate objects in your response. In that case, also generate
a clarification question that clearly distinguishes between the candidates.

Do NOT include object IDs in the clarification question!

Respond ONLY with a JSON object in the following format:
{
    "explanation": "Briefly explain why each object ID is or is not identified",
    "object": "Expression of the target object mentioned by the user"
    "object_ids": [<object_id_1>, <object_id_2>, ...],
    "clarification_question": "A question to help resolve the ambiguity between the
    candidates"
}

Now, based on the dialogue, the object list, and the images below, provide your response.
```

Figure 6: End-to-end baseline prompt for Agent.

### C.2 IMAGE-GROUNDED SCENE GRAPH SYSTEM

For the proposed structured prompting method, we provide the following prompts:

- Dialogue analysis:
  - User answer analysis (Agent): Fig. 7
  - Agent question analysis (User): Fig. 9
- Reference resolution:
  - Object reference extraction: Fig. 8

- Relevant object types extraction: Fig. 10
- Information specification: Fig. 11
- Target object prediction: Fig. 12
- Captioning:
  - Object description generation (User): Fig. 13
  - Ambiguous object description (User): Fig. 14
  - Closed clarification questions (Agent): Fig. 15
- Open clarification question (Agent): Fig. 16
- Answer formulation (User): Fig. 17

```
You are a helpful assistant specialized in analyzing dialogues.

I will provide a dialogue between an user and an agent, in which the user attempts to
locate an object within a scene with the help of the agent. Your task is to predict
the next action the agent should take.

Possible Actions:
1. Conclude the dialogue and predict the target object's ID.
2. Perform reference resolution based on the user's response.

Respond ONLY with a JSON object in the following format:
{
    "action": 1 or 2,
    "target_object_id": "<object_id>" or null
}

Examples:
1.
Dialogue:
- User: Find the brown chair next to the desk.

Output:
{
    "action": 2,
    "target_object_id": null
}

2.
Dialogue:
- User: Look for the couch in the corner of the room.
- Agent: I see two couches in the corner of the room. Do you mean the grey one or
the brown one? (the grey one: 2, the brown one: 5)
- User: It's the brown one.

Output:
{
    "action": 1,
    "target_object_id": 5
}

3.
Dialogue:
- User: Your target is the orange cup.
- Agent: I found two orange cups. Do you mean the cup on the table or the cup on
the shelf? (the cup on the table: 13, the cup on the shelf: 17)
- User: Neither, it is the red one next to the microwave.

Output:
{
    "action": 2,
    "target_object_id": null
}
```

Figure 7: Dialogue analysis prompt for Agent (1/2).

```
4.
Dialogue:
- User: Look at the window.
- Agent: I can see three windows, which window do you mean?
- User: The smallest one on the left.

Output:
{
    "action": 2,
    "target_object_id": null
}

Now, based on the dialogue below, provide your response.
```

Figure 7: Dialogue analysis prompt for Agent (2/2).

```
You are a helpful assistant specialized in 3D visual grounding.

I will provide a dialogue between an user and an agent, in which the user attempts
to locate an object within a scene with the help of the agent. Your task is to extract
the target object(s) referred by the agent/user.

Respond ONLY with a Python list of target objects.

Examples:
1.
Dialogue:
- User: Find the brown chair next to the desk.

Output:
["the brown chair next to the desk"]

2.
Dialogue:
- User: Look for the couch in the corner of the room.
- Agent: I see two couches in the corner of the room. Do you mean the grey one or the
brown one?

Output:
["the grey couch in the corner of the room", "the grey couch in the corner of the room"]

3.
Dialogue:
- User: Your target is the orange cup.
- Agent: I found two orange cups. Do you mean the cup on the table or the cup on
the shelf?
- User: Neither, it is the red one next to the microwave.

Output:
["the red cup next to the microwave"]

4.
Dialogue:
- User: Look at the window.
- Agent: I can see three windows, which window do you mean?
- User: The smallest one on the left.

Output:
["the smallest window on the left"]

Now, based on the dialogue below, provide your response.
```

Figure 8: Object reference extraction prompt for Agent.

```
You are a helpful assistant specialized in analyzing dialogues.

I will provide a dialogue between a user and an agent, in which the agent attempts
to identify a object within a scene that is only known to the user. Your task is
to identify the type of the agent's most recent clarification question.

Question Types:
1. Open question
2. Closed question

Respond ONLY with 1 or 2.

Examples:
1.
Dialogue:
- Agent: Which object do you have in mind?

Output:
1

2.
Dialogue:
- Agent: Which object do you have in mind?
- User: the window.
- Agent: I can see three windows, which window do you mean?

Output:
1

3.
Dialogue:
- Agent: Which object do you have in mind?
- User: the couch in the corner of the room.
- Agent: I see two couches in the corner of the room. Do you mean the grey one or
the brown one?

Output:
2

4.
Dialogue:
- Agent: Which object do you have in mind?
- User: Find the orange cup.
- Agent: I see two orange cups. One is on the dining table, and the other one is on
the shelf, which one do you mean?

Output:
2

Now, based on the dialogue below, provide your response.
```

Figure 9: Dialogue analysis for User.

```
You are a helpful assistant specialized in 3D visual grounding.

I will provide a list of object classes in a scene. Your task is to identify those
relevant to an utterance.

Respond ONLY with a Python list of selected object classes.

Now, based on the utterance and the object list below, provide your response.
```

Figure 10: Prompt.

```
You are a helpful assistant specialized in 3D visual grounding.

I will provide an utterance that refers to an object in a scene. Your task is to
identify the target object's ID from a list of objects based on the utterance.

You should determine the relevant information and the associated object IDs that
support your decision.

Available Information:
(a) Object Image: Provides visual details (e.g., color, shape, material, state, etc.).
(b) Size: Indicates whether an object is big or small.
(c) Dimensions: Helps differentiate objects as long/short, wide/narrow, or tall/short.
(d) Distance Between Objects: Indicates whether objects are near or far.
(e) Multi-Object Image: Shows spatial relationships between objects (e.g. on, above,
under, beside, left/right,
between, front, behind, etc.).
(f) Top-Down Scene View: Provides scene layout (e.g. wall, corner or middle of the room).

Respond ONLY with a JSON object in the following format:
{
    "(a-f)": {
        "object_ids": [<object_id_1>, <object_id_2>, ...],
        "usage": "Use <information_item> to ..."
    },
    ...
}

Now, based on the utterance and the object list below, provide your response.
```

Figure 11: Information specification prompt for Agent.

```
You are a helpful assistant specialized in 3D visual grounding.

I will provide an utterance that refers to an object in a scene. Your task is to
identify the target object's ID from a list of objects, based on the utterance and
the provided information.

***Important Instruction***
- There may be more than one candidate object that likely satisfies the conditions
described in the utterance.
Include ALL such candidate objects in your response.
- If no candidate object satisfies the conditions, return an empty list.

Respond ONLY with a JSON object in the following format:
{
    "explanation": "Briefly explain why each object ID is or is not identified,
    referring to the provided information",
    "target_object_ids": [<object_id_1>, <object_id_2>, ...]
}

Now, based on the utterance, the object list, and the information below, provide
your response.
```

Figure 12: Target object prediction prompt for Agent.

```
You are a helpful assistant specialized in generating referring expressions for 3D scenes.

I will provide a target object and distractor objects within a scene that all satisfy
an utterance. Your task is to generate a referring expression that clearly identifies
the target object, distinguishing it from the distractor objects.

You will also have access to other objects in the scene, which can be used as anchors
to help refer to the target object.

Available Information:
(a) Object Image: Provides visual details (e.g., color, shape, material, state, etc.).
(b) Size: Indicates whether an object is big or small.
(c) Dimensions: Helps differentiate objects as long/short, wide/narrow, or tall/short.
(d) Distance Between Objects: Indicates whether objects are near or far.
(e) Multi-Object Image: Shows spatial relationships between objects (e.g. on, above,
under, beside, left/right,
between, front, behind, etc.).
(f) Top-Down Scene View: Provides scene layout (e.g. wall, corner or middle of the room).

Important Instruction:
- NEVER include object IDs in the expression!
- Do NOT use unlisted anchor objects.
- Do NOT repeat information already existing in the utterance.
- Use as little information as possible.
- Use as few anchor objects as possible.
- Do NOT use ambiguous anchor objects.
- When using (b), (c), or (d), ensure the differences of these numbers between the
objects are VERY substantial.
- If using left/right orientation, you MUST include a clear anchor object for reference.

Respond ONLY with a JSON object in the following format:
{
    "explanation": "Explain why the information is selected (by referring to numbers
    if used)",
    "expression": "referring expression to the target object"
}

Examples:
1.
Utterance:
Find the chair.

> Output:
{
    "explanation": "...",
    "expression": "It is the brown chair next to the desk."
}

2.
Utterance:
Find the cup on the dining table.

> Output:
{
    "explanation": "...",
    "expression": "It's the largest one."
}
```

Figure 13: Object description generation prompt (1/2).

```
3.
Utterance:
Find the cabinet by the door.

> Output:
{
    "explanation": "...",
    "expression": "It's the one by the open door."
}

4.
Utterance:
Find the bookshelf.

> Output
{
    "explanation": "...",
    "expression": "It is the bookshelf on the left of the window."
}

5.
Utterance:
Find the grey trash can.

> Output:
{
    "explanation": "...",
    "expression": "It's the one with a white bag."
}

6.
Utterance:
Find the desk with a monitor on it.

> Output:
{
    "explanation": "...",
    "expression": "It's the one in the middle of the room."
}

Now, based on the utterance, the target object, the distractor objects, other objects,
and the information below, generate your referring expression.
```

Figure 13: Object description generation prompt (2/2).

```
You are a helpful assistant specialized in generating referring expressions for 3D
scenes.

I will provide multiple objects within a scene. Your task is to generate an ambiguous
referring expression that applies to all of these objects.

You will also have access to other objects in the scene, which can be used as anchors
to help refer to the objects.

Available Information:
(a) Object Image: Provides visual details (e.g., color, shape, material, state, etc.).
(b) Size: Indicates whether an object is big or small.
(c) Dimensions: Helps differentiate objects as long/short, wide/narrow, or tall/short.
(d) Distance Between Objects: Indicates whether objects are near or far.
(e) Multi-Object Image: Shows spatial relationships between objects (e.g. on, above,
under, beside, left/right, between, front, behind, etc.).
(f) Top-Down Scene View: Provides scene layout (e.g. wall, corner or middle of the room).

Important Instruction:
- NEVER include object IDs in the expression!
- Do NOT use unlisted anchor objects.
- Ambiguity can arise from either the target objects or the anchor objects.
- When using (b), (c), or (d), ensure the differences of these numbers between the
objects are VERY small.

Respond ONLY with a JSON object in the following format:
{
    "explanation": "Explain the ambiguity in the expression (by referring to numbers
    if used)",
    "expression": "referring expression to the objects"
}

Examples:
1.
{
    "explanation": "...",
    "expression": "It is the brown chair."
}

2.
{
    "explanation": "...",
    "expression": "Look at the small cup."
}

3.
{
    "explanation": "...",
    "expression": "Find the cabinet by the door."
}

4.
{
    "explanation": "...",
    "expression": "Your target is the window above the kitchen counter."
}

5.
{
    "explanation": "...",
    "expression": "It's the grey trash can with a white bag."
}
```

Figure 14: Ambiguous object description generation prompt (1/2).

```
6.
{
    "explanation": "...",
    "expression": "Find the desk in the corner of the room."
}

Now, based on the ambiguous objects, other objects, and the information below, generate
your referring expression.
```

Figure 14: Ambiguous object description generation prompt (2/2).

```
You are a helpful assistant specialized in generating clarification questions for
3D scenes.

I will provide two objects within a scene that both satisfy an utterance. Your task is
to generate a clarification question that clearly distinguishes between them based on
the provided information.

You will also have access to other objects in the scene, which can be used as anchors
to help refer to the objects to be clarified.

Available Information:
(a) Object Image: Provides visual details (e.g., color, shape, material, state, etc.).
(b) Size: Indicates whether an object is big or small.
(c) Dimensions: Helps differentiate objects as long/short, wide/narrow, or tall/short.
(d) Distance Between Objects: Indicates whether objects are near or far.
(e) Multi-Object Image: Shows spatial relationships between objects (e.g. on, above,
under, beside, left/right, between, front, behind, etc.).
(f) Top-Down Scene View: Provides scene layout (e.g. wall, corner or middle of the room).

You can generate either:
> a yes-no question by only referring to one of the candidate objects, or
> a two-option question by referring to both candidate objects,
whichever is more natural.

Important Instruction:
- NEVER include object IDs in the question!
- Do NOT use unlisted anchor objects.
- Use as little information as possible.
- Use as few anchor objects as possible.
- Do NOT use ambiguous anchor objects.
- When using (b), (c), or (d), ensure the differences of these numbers between the
objects are VERY substantial.
- If using left/right orientation, you MUST include a clear anchor object for reference.

Respond ONLY with a JSON object in the following format:
{
    "explanation": "Explain why the information is selected (by referring to numbers
    if used)",
    "<object_1_id>": "referring expression of the object",
    "<object_2_id>": "referring expression of the object",
    "question": "clarification question"
}

Examples:
1.
- Utterance:
Find the trash can.

> Output:
{
    "explanation": "...",
    "4": "the one by the door",
    "6": "",
    "question": "I found two trash cans. Do you mean the one by the door?"
}
```

Figure 15: Closed clarification question generation prompt.

```
2.
Utterance:
Find the window above the kitchen counter.

> Output:
{
    "explanation": "...",
    "23": "the bigger one",
    "24": "the smaller one",
    "question": "I can find two windows above the kitchen counter, a bigger one and a
    smaller one. Which one do you mean?"
}

3.
Utterance:
Find the wooden table next to the couch.

> Output:
{
    "explanation": "...",
    "23": "the one next to the black couch",
    "24": "the one next to the red couch",
    "question": "I can see two wooden tables next to a couch. Do you mean the one next
    to the black couch or the one next to the red couch?"
}

Further Question Examples:
4. <ambiguity summarization>. Do you mean the one near the open door or the one near
the closed door?
5. <ambiguity summarization>. Do you mean the taller one?
6. <ambiguity summarization>. Do you mean the one on the desk?
7. <ambiguity summarization>. Do you mean the one on the left of the window or the one
on right?
8. <ambiguity summarization>. Do you mean the one in the middle of the room or the one
in the corner?

Now, based on the utterance, the objects to clarify, other objects, and the information
below, generate your clarification question.
```

Figure 15: Closed clarification question generation prompt.

```
You are a helpful assistant specialized in generating clarification questions for
3D scenes.

I will provide a result of reference resolution, which includes an utterance describing
the target object and multiple candidate object IDs. Your task is to generate a question
that asks for clarification about the intended target object.

Examples:
1.
Result:
{
    "object": "Find the trash can.",
    "object_ids": [3, 6, 12]
}

> Output:
I can see three trash cans, which one do you mean?

2.
Utterance:
Result:
{
    "object": "Find the window above the kitchen counter.",
    "object_ids": [21, 22, 26, 28]
}

> Output:
I found four windows above the kitchen counter, which window do you mean?

3.
Utterance:
{
    "object": "Find the wooden table next to the couch.",
    "object_ids": []
}

> Output:
I can't find any wooden tables next to the couch, which one do you mean?

Now, based on the result below, generate your clarification question.
```

Figure 16: Open clarification question generation prompt for Agent.

```
You are a helpful assistant specialized in generating responses to clarification
questions.

I will provide a dialogue between a user and an agent, in which the agent attempts to
identify a object within a scene that is only known to the user. Your task is to
formulate an answer to the agent's clarification question, based on the reference
resolution result and the target object's ID in mind.

Examples:
1.
Dialogue:
- User: Look for the couch in the corner of the room.
- Agent: I see two couches in the corner of the room. Do you mean the grey one or the
brown one?

Result:
{
    "the grey couch in the corner of the room": 12,
    "the brown couch in the corner of the room": 16
}

Target Object ID:
12

Output:
It is the grey one.

2.
Dialogue:
- User: Your target is the orange cup.
- Agent: I found two orange cups. Do you mean the cup on the table or the cup on the
shelf?

Result:
{
    "the orange cup on the table": 3,
    "the orange cup on the shelf": 7,
}

Target Object ID:
7

Output:
The cup on the shelf.

Now, based on the dialogue, the reference resolution result, and the target object's ID
below, generate your answer.
```

Figure 17: Answer formulation prompt for User.

