# OpenReview forum: "Interactive Object Grounding Using Image-Grounded Scene Graphs and Prompt Chaining"
_ICLR.cc/2026/Conference — ICLR 2026 Conference Withdrawn Submission_

### Official Review · Reviewer_yDiR · 2025-10-28

**Soundness:** 3
**Presentation:** 3
**Contribution:** 2
**Rating:** 4
**Confidence:** 4

**Summary:**

This paper proposes an Interactive Object Grounding (IOG) framework that combines a novel Image-Grounded Scene Graph (IGSG) with a prompt chaining strategy to perform multimodal reference resolution and clarification dialogue in 3D environments. The method integrates both LLM and VLM modules for object identification and clarification question generation. Unlike end-to-end visual grounding approaches that require numerous images, IGSG dynamically samples relevant multimodal information, thus reducing computational cost while improving interpretability and dialogue quality.

**Strengths:**

1. Defines Interactive Object Grounding, bridging vision-language grounding with interactive dialogue.

2. Introduces IGSG, a lightweight yet expressive multimodal representation that supports dynamic image selection and zero-shot multimodal reasoning via prompt chaining.

**Weaknesses:**

1. This work mainly applies existing models to a new robotics application. However, the scenario does not appear to present significant challenges, and the proposed solution seems to be a straightforward extension to this setting.

2. Only 8 participants were involved, limiting statistical significance and ecological validity.

3. The contribution of each module (IGSG, prompt chaining, clarification strategy) is not individually quantified.

4. The multi-step prompting pipeline, while elegant, may be difficult to scale to real-time robotic applications.

5. The work does not include comparisons with recent multimodal dialogue systems such as SIMMC or clarification-based interactive grounding datasets (e.g., IGLU, PhotoBook).

**Questions:**

Refer to the weaknesses above.

---

> ### Author Response · Authors · 2025-11-25
> **Rebuttal**
>
> We thank the reviewer for their comments and would like to clarify the following points.
>
> *Scenario/Task*
>
> First, we would like to point out that the task of interactive object grounding presents major challenges.  In addition to the core task of multimodal reference resolution, it requires object captioning for the purpose of clarification question generation, a new task in itself, as well as dialogue understanding and generation.  All these subtasks are challenging in themselves in the context of the complex 3D scenes in ScanNet; combining them into an effective conversational system only adds to that.  As we were aiming for a zero-shot solution, we used existing foundation models to drive the prompt chaining process.
>
> *Evaluation*
>
> We are not sure what the reviewer means by the IGSG, prompt chaining, and clarification strategy not being ‘individually quantified’.  In our experiments, we are directly evaluating the clarification strategy by comparing performance before and after clarification.  The combination of the Image-Grounded Scene Graph with prompt chaining is what we propose as the methodology underlying all of the components.  They cannot be evaluated individually.  Evaluating individual components such as captioning would require major annotation efforts.
>
> We have considered other benchmarks such as SIMMC, IGLU, and Photobook, but eventually dismissed them as unsuitable for our purposes.  Both Photobook and SIMMC only provide 2D images in isolation and therefore do not sufficiently represent our target scenario of complex 3D scenes.  IGLU does provide 3D scenes, but these are very artificial blocks world style scenes for which an image-grounded scene graph would not make much sense.  ScanNet on the other hand provides complex 3D scenes based on collections of real-world images, along with point cloud information to create scene graphs.
>
> We agree that that the scale of the human evaluation is currently very limited and would not present very strong evidence in favour of the proposed method on its own.  However, combined with the results obtained with the simulated user we believe it makes the overall evaluation stronger.
>
> *Scalability*
>
> Multi-step prompting of large foundation models might generate too much latency for practical application of our method in its current form.  However, single prompt alternatives might require very large prompts, which will also create more latency.  We leave addressing latency issues, for example by creating smaller, more specialised models through fine-tuning or distillation, for future work.

---

> > ### Comment · Reviewer_yDiR · 2025-11-28
> >
> > Thanks for the response. I have read the rebuttal carefully and appreciate the clarifications provided. Some concerns have been addressed, but several remain unresolved, as follows:
> >
> > 1. Component Analysis (**Main Concern**): The explanation for the absence of ablation studies still feels somewhat insufficient. The authors note that IGSG and prompt chaining “cannot be evaluated individually” and point to annotation costs as the reason. However, to more comprehensively validate the method, it would still be helpful to distinguish, at least to some extent, the respective contributions of the structured visual representation (IGSG) and the inference strategy (prompt chaining). Comparisons such as “Text-only Scene Graph + Prompt Chaining” or “IGSG + Single Prompt” should be feasible in practice and would provide a clearer picture of where the improvements originate. Without such analyses, it remains difficult to determine whether the gains primarily stem from the graph construction or from the task decomposition itself.
> >
> > 2. Regarding Human Evaluation: As the authors acknowledged, the scale of the human evaluation is very limited. While I understand the difficulty of such experiments, in the absence of strong ablation studies on the automated side, the lack of a more extensive human evaluation makes it difficult to verify the robustness of the system in broader scenarios.
> >
> > While the problem setting is interesting and the clarification mechanism shows promise, the lack of rigorous component analysis (ablation studies) and the limited scale of human evaluation limit the paper's contribution and validation. Therefore, I would prefer to maintain my previous score (4).

---

> ### Author Response · Authors · 2025-12-03
>
> Our explanation of the E2E baseline may have been somewhat inaccurate by implying it does not use an Image-Grounded Scene Graph.  The reviewer’s suggested “IGSG + Single Prompt” system variant corresponds to the E2E variant we have evaluated.  We have now also carried out two additional ablations, related to the "Text-only Scene Graph + Prompt Chaining" variant mentioned by the reviewer.  For the first ablation we removed the use of images from the reference resolution module ("IGSG txt RefRes"), and for the second ablation we additionally removed the use of images from the captioning module ("IGSG txt RefRes Capt").  In both cases, the simulated user was left unchanged, starting every dialogue with an ambiguous answer referring only to the target object type (the ambTp-IGSG condition in the paper).
>
> The results in the table below show that the performance levels before clarification are very similar.  This is to be expected, because the initial reference resolution step is based on the given object type only, so the candidates can be selected directly from the object list without the need to refer to any images.  The performance levels after clarification are lower without access to the images, but only moderately, because the agent can 'control the narrative' by asking closed clarification questions based on text information only.  The simulated user can answer such questions by choosing between the options and then the agent can resolve the ambiguity in a straightforward manner without needing any images.  On the other hand, after an open clarification question, the user may refer to images to describe the target object, causing a problem for the agent to then perform reference resolution without access to any images.
>
> |  Condition                     |   P-before  | R-before  | F-before | P-after | R-after  | F-after  | F-gain |
> | ------------------------------ | -------- | ------- | -------- | -------- | -------- | -------- | -------- |
> | IGSG                            | 40.8   | 98.9  | 57.7  |  86.5  |  80.6  |  83.5  |  +25.8  |
> | IGSG txt RefRes          | 41.5   | 98.1  | 58.4  |  82.0  |  77.9  |  79.9  |  +21.5 |
> | IGSG txt RefRes Capt  |  40.8   | 97.0  | 57.5  |  84.1  |  79.9  |  81.9  |  +24.4 |

---

### Official Review · Reviewer_GA34 · 2025-10-29

**Soundness:** 2
**Presentation:** 1
**Contribution:** 1
**Rating:** 2
**Confidence:** 4

**Summary:**

The paper proposes a task called Interactive Object Grounding, which is similar to conventional visual grounding but requires the agent to follow up and ask clarifying questions when the description is ambiguous and may not refer to a single object. To address this task, the authors propose a system that utilizes prompt chaining over an Image-Grounded Scene Graph (IGSG). A benchmark is also provided, built on ScanNet, and designed for ambiguous same-type objects (65 scenes, 263 tasks, with an average of 1.33 distractors per task).

**Strengths:**

The proposed task and problem is useful and interesting.

**Weaknesses:**

1. The paper introduces a new task, a method, and a benchmark, but none of them are clearly or thoroughly explained.

2. The authors do not provide sufficient comparison or discussion of existing work in single- or multi-object grounding. While the proposed setting is indeed different and direct comparison may not be entirely fair, some level of quantitative or qualitative comparison and discussion is still necessary to position their contribution.

3. There is also limited quantitative evaluation of existing methods to support their claim that previous approaches “generate lengthy and sometimes confusing clarification questions.” Although the claim is reasonable, the paper does not demonstrate how severe the issue is or how much their method improves upon it. Beside it seems that only GPT-4.1 is tested here.

4. The average number of distractors is only 1.33, which makes the task relatively simple and may not effectively test the model’s ability to disambiguate the target in more complex or cluttered scenes.

**Questions:**

Please address the weakness mentioned above.

---

> ### Author Response · Authors · 2025-11-25
> **Rebuttal**
>
> We thank the reviewer for their comments.
>
> To address their concerns about our presentation of the task, method, and benchmark, we would ask the reviewer to indicate more specifically what is unclear.  We have tried to describe the interactive grounding task, image-grounded scene graph with prompt chaining method, and the experiments, in as much detail as possible within the given page limit.  We also provided a discussion of related work in the introduction and related work sections.  Could the reviewer please be more specific about this as well?
>
> Our observation that recent foundation models “generate lengthy and sometimes confusing clarification questions” was intended to informally describe the limitations of these models when used out-of-the-box, in end-to-end fashion.  The proposed scene-graph based prompt chaining method attempts to address this and is evaluated in terms of object grounding performance.
>
> Our paper does indeed currently only report results with gpt4.1.  We have since carried out some experiments with the smaller, open-source, Qwen3-VL-30B-A3B-Instruct model.  In the table below, we report Precision, Recall, and F-score before and after clarification, as well as the absolute gain in F-score.  The results show that the performance levels with Qwen are lower, as expected, but also the same pattern of improvements after clarification and superior performance of IGSG over E2E.
>
> |  Condition                  |  P-before  | R-before  | F-before | P-after | R-after  | F-after  | F-gain |
> | ---------------------------- | -------- | ------- | -------- | -------- | -------- | -------- | -------- |
> | ambTp-E2E-GPT      | 42.1   | 100   | 59.3  |  88.3  |  89.0  |  88.6  |  +29.3  |
> | ambTp-E2E-Qwen    | 39.1   | 87.1  | 53.9  |  66.5  |  69.6  |  68.0  |  +14.1 |
> | ambTp-IGSG-GPT    | 41.5  |  99.6  | 58.6  |  92.8  |  92.8  |  92.8  |  +34.2  |
> | ambTp-IGSG-Qwen  | 40.8   | 98.9  | 57.7  |  86.5  |  80.6  |  83.5  |  +25.8 |
>
> To provide further insight into how the complexity of the task in terms of number of distractors might affect the results, we have now included results for the cases with one distractor only.  We also doubled the size of the dataset by interchanging target and distractor objects.  The table below shows that the results for the extended dataset are very similar to those of the original dataset.  As expected, when only considering the cases with one distractor, the scores before clarification are higher and the gains through clarification are smaller, especially in the E2E setting (+9.2 vs +13.2).  More importantly, gains made with the IGSG system are much bigger than with the E2E system, regardless of whether only the 1 distractor cases are considered or not.
>
> | Dataset	      | Condition                 | P-before        | R-before       | F-before       | P-after	| R-after	| F-after	| F-gain |
> | ---------------- | -------------------------- | -------- | ------- | ------- | -------- | ------- | ------ | --------- |
> | Original	      | ambTp-E2E-Qwen	| 39.1    | 87.1	| 53.9   | 66.5	| 69.6   | 68.0	| +14.1 |
> | Extended     | ambTp-E2E-Qwen	| 39.2    | 87.1	| 54.0   | 65.8	| 68.6   | 67.2	| +13.2 |
> | 1 distractor  | ambTp-E2E-Qwen	| 45.2    | 90.9	| 60.4   | 68.4	| 71.0   | 69.6	| +9.2   |
> | Original       | ambTp-IGSG-Qwen | 40.8    | 98.9	| 57.7   | 86.5	| 80.6   | 83.5	| +25.8 |
> | Extended    | ambTp-IGSG-Qwen | 40.4    | 98.9	| 57.3   | 86.8	| 81.4   | 84.0	| +26.7 |
> | 1 distractor | ambTp-IGSG-Qwen	| 47.0    | 99.5	| 63.9   | 90.9	| 87.9   | 89.3	| +25.4 |

---

> ### Comment · Reviewer_GA34 · 2025-11-26
>
> Thank you for the rebuttal and the detailed clarifications. My main remaining concerns focus on the system setup, particularly how future work is expected to engage with the simulated user framework, as well as the limited quantitative analysis provided to support the core claims.
> 1. Clarification of task formulation and system architecture
>
> A clearer formalization of the task would be helpful. My understanding is that the goal is to build an agent capable of grounding target objects in 3D scenes while interacting with a user and generating clarification questions to disambiguate ambiguous descriptions. If this is correct, I would ask the authors to clarify:
>
> - Is the simulated user an essential component of the system, or only an evaluation utility?
> - Since all components in this work are implemented via GPT-4.1, are future systems expected to use the same model for every sub-module, or can components be different?
>
> Additionally, the performance of the agent appears highly dependent on the correctness of responses generated by the simulated user. However, are they truly able to provide precise and human-like descriptions? It would strengthen the contribution to validate the simulated user’s captioning capability on 3D dense captioning benchmarks [1]. Moreover, does the agent itself truly possess the ability to identify targets in 3D space? Evaluating the agent independently using established single- or multi-object grounding datasets [2,3, 4] would help demonstrate its actual grounding ability.
>
> In general i think the idea is interesting, but several claims currently lack quantitative grounding.
>
> ---
>
> 2. Dataset statistics and analysis
> More detailed dataset analysis would make the contribution clearer. such as vocabulary diversity, distribution of object categories etc. Without such analysis, it is difficult to fully assess dataset difficulty, diversity, and contribution.
>
> Additionally, many figures require cross-referencing the text to understand. More descriptive captions explaining the flow, decisions, and purpose of each figure would improve readability.
>
> [1] Chen, Zhenyu, et al. "Scan2cap: Context-aware dense captioning in rgb-d scans." Proceedings of the IEEE/CVF conference on computer vision and pattern recognition. 2021.
>
> [2] Achlioptas, Panos, et al. "Referit3d: Neural listeners for fine-grained 3d object identification in real-world scenes." European conference on computer vision. Cham: Springer International Publishing, 2020.
>
> [3] Chen, Dave Zhenyu, Angel X. Chang, and Matthias Nießner. "Scanrefer: 3d object localization in rgb-d scans using natural language." European conference on computer vision. Cham: Springer International Publishing, 2020.
>
> [4] Zhang, Yiming, ZeMing Gong, and Angel X. Chang. "Multi3drefer: Grounding text description to multiple 3d objects." Proceedings of the IEEE/CVF International Conference on Computer Vision. 2023.

---

> > ### Author Response · Authors · 2025-12-03
> >
> > The reviewer's understanding of the task is correct. Given a 3D environment perceived by both the agent and a user, the task for the agent is to identify a specific target object by asking clarification questions to the user, who knows the target object.  The task for the (human or simulated) user is to answer the agent’s questions, given the target object.  After every question-answer exchange, the agent makes a prediction in the form of a list of candidate target object hypotheses, based on the conversation so far.  The grounding performance is measured in terms of precision, recall, and F-score, with the target object as reference.
> >
> > The simulated user is introduced to evaluate the agent’s ability to perform this task without requiring a human user.  Ultimately, the agent is intended to interact with human end-users, so the simulated user should generate realistic responses for the evaluation to be meaningful.  One could evaluate this ability by having human judges rate/compare simulated responses, but that is a costly process, and arguably less crucial as we have also evaluated the agent directly with human users.
> >
> > We specify for each sub-module which foundation model to use, so the overall system could rely on a mixture of different models (in fact, we have used gpt4o for determining the parameters for generating top-down scene views, and gpt4.1 for all other LLM/VLM tasks).
> >
> > Our contributions are focused primarily on the interactive grounding task, with our reference resolution and captioning modules being specifically designed for handling clarification dialogue.  To evaluate these modules on standard benchmarks we would have to make them suitable for the tasks underlying these benchmarks (one-shot, non-conversational), thus making the evaluation less relevant to our claims.  Individually evaluating these modules on the conversational data that we collected would require a significant annotation effort.  We will reformulate the presentation of our contributions to avoid confusion about the core claims.

---

### Official Review · Reviewer_17AJ · 2025-11-01

**Soundness:** 2
**Presentation:** 2
**Contribution:** 3
**Rating:** 4
**Confidence:** 4

**Summary:**

This paper addresses the task of interactive object grounding, where a system must link a user's linguistic expression to an object in a 3D scene and ask for clarification if the expression is ambiguous. The authors note that end-to-end VLM-based systems can be costly and produce poor clarification questions, while text-only scene graph methods lack rich visual detail. They propose a modular, prompt-chaining approach centered on an "Image-Grounded Scene Graph" (IGSG). This IGSG stores basic object properties and maps objects to relevant images, allowing the system to dynamically sample and feed specific multimodal information to LLM/VLM modules for reference resolution and clarification question generation. Evaluations on a new benchmark derived from ScanNet show the IGSG method outperforms an end-to-end baseline in grounding accuracy, particularly after clarification, and at a significantly reduced computational cost.

**Strengths:**

1. The core idea of the Image-Grounded Scene Graph (IGSG) is an intuitive and practical compromise. It avoids the high cost of feeding all scene images to a VLM while retaining richer visual grounding than purely symbolic scene graphs.
2. The performance gains in terms of computational cost (reportedly 35% of the cost of the baseline) could be practically relevant for real-world applications.
3. The paper is well-written and easy to follow.

**Weaknesses:**

1. The primary baseline (E2E) feels insufficient. It is not surprising that a single, general-purpose VLM (GPT-4.1) prompted end-to-end performs poorly on this specialized, multi-turn task compared to a purpose-built, modular pipeline that uses the same model for its components. A stronger baseline might have involved other 3D grounding methods adapted for this interactive task, or a fine-tuned model.
2. The evaluation with human subjects, while a good inclusion, is very small (8 subjects from within the authors' organization). This small sample size and potential for in-group bias limit the conclusiveness of these findings. The paper also notes that "human users struggled to grasp the complex 3D scenes," which may also suggest issues with the user interface or task setup.
3. The method relies on "gpt-4.1" (presumably a specific version of GPT-4), a powerful, closed-source model. This makes exact replication difficult and ties the method's success to a specific proprietary API, limiting insights into how it would perform with other models.

**Questions:**

1. How dependent is the IGSG method's success on the specific VLM used? Have the authors experimented with other open-source models for the VLM-dependent modules?
2. The IGSG requires pre-processing of the scene (point cloud, segmentation, mapping images to objects). What is the computational overhead of creating the IGSG for a new scene, and how does this pre-processing cost compare to the per-dialogue savings?

---

> ### Author Response · Authors · 2025-11-25
> **Rebuttal**
>
> We thank the reviewer for appreciating the merits of our approach and thoughtful comments.
>
> *Baselines*
>
> As the reviewer suggests, one might expect that a scene graph based modular system performs better than an E2E baseline on this complex multimodal dialogue task.  However, a modular approach might also introduce limitations/bottlenecks that an E2E system does not suffer from, so that expectation is not completely obvious.   Furthermore, we believe that the advantage of cost savings through efficient use of images through the scene graph is at least as important.
> We also would like to point out that our method does not use any fine-tuning, so comparisons with other zero-shot baselines such as the E2E variant we created would make the most sense.  Given the fact that any additional baseline would require a significant amount of effort to fit into our specific task, we focused on evaluating the effectiveness of our clarification method and the comparison with the E2E approach.  Other 3D grounding methods based on scene graphs, such as Transcribe3D  [1] and SeeGround  [2] do not support object captioning and can therefore not be directly compared with our approach.
>
> *Human evaluation*
>
> We agree that that the scale of the human evaluation is currently very limited and would not present very strong evidence in favour of the proposed method on its own.  However, combined with the results obtained with the simulated user we believe it makes the overall evaluation stronger.  Because accommodating larger scale evaluations with real users requires significant further engineering effort, we had to leave that for future work.
>
> Regarding the experimental design of the human evaluation, the task was indeed quite challenging for the user, which explains the difference in performance levels with the results from the simulated users.  However, our main goal was to demonstrate the effectiveness of our proposed clarification method, regardless of how challenging it was for the users in general.
>
> *Foundation model*
>
> To address the reviewer’s concern about the foundation model used, we have repeated some of the experiments, replacing the gpt4.1 model with the smaller, open-source, Qwen3-VL-30B-A3B-Instruct model.  In the table below, we report Precision, Recall, and F-score before and after clarification, as well as the absolute gain in F-score.  As expected, the performance levels overall are lower when using the Qwen model, but we also observe the same pattern of improvements after clarification and superior performance of IGSG over E2E.
>
> |  Condition                  |  P-before  | R-before  | F-before | P-after | R-after  | F-after  | F-gain |
> | ---------------------------- | -------- | ------- | -------- | -------- | -------- | -------- | -------- |
> | ambTp-E2E-GPT      | 42.1   | 100   | 59.3  |  88.3  |  89.0  |  88.6  |  +29.3  |
> | ambTp-E2E-Qwen    | 39.1   | 87.1  | 53.9  |  66.5  |  69.6  |  68.0  |  +14.1 |
> | ambTp-IGSG-GPT    | 41.5  |  99.6  | 58.6  |  92.8  |  92.8  |  92.8  |  +34.2  |
> | ambTp-IGSG-Qwen  | 40.8   | 98.9  | 57.7  |  86.5  |  80.6  |  83.5  |  +25.8 |
>
> *References*
>
> 1. Fang, Jiading, et al. "Transcribe3d: 3d referring expression resolution through large language models." 2024 IEEE/RSJ International Conference on Intelligent Robots and Systems (IROS). IEEE, 2024.
> 2. Li, Rong, et al. "Seeground: See and ground for zero-shot open-vocabulary 3d visual grounding." Proceedings of the Computer Vision and Pattern Recognition Conference. 2025.

---

### Official Review · Reviewer_ysk7 · 2025-11-01

**Soundness:** 2
**Presentation:** 3
**Contribution:** 2
**Rating:** 6
**Confidence:** 4

**Summary:**

This paper explores interactive object grounding in 3D scenes. It proposes an Image-Grounded Scene Graph (IGSG) framework that dynamically selects relevant visual information through prompt chaining, enabling effective collaboration between LLMs and VLMs. Evaluated on a subset of ScanNet with distractors, the proposed method significantly improves clarification and grounding accuracy compared to an end-to-end baseline.

**Strengths:**

1. The motivation is solid and well-justified, effectively grounding ambiguous natural language instructions in complex 3D environments is a key challenge for embodied AI.
2. The proposed Image-Grounded Scene Graph (IGSG) offers a dynamic approach to integrating multimodal information via prompt chaining, allowing LLMs and VLMs to collaborate effectively without relying on a rigid symbolic scene graph.
3. The method achieves strong performance while maintaining a significantly lower computational cost compared to e2e baseline.

**Weaknesses:**

1. The newly formatted dataset with 263 instances is relatively small, which weakens the strength of the paper’s experimental statement.
2. Error propagation during interaction is not clearly discussed. What happens if the predicted object ID list is incorrect at the beginning of the dialogue?
3. The comparative analysis is limited, as the method is only compared against an e2e baseline. It would be more convincing to include additional baselines, such as the text-SG model mentioned in the paper.

**Questions:**

1. It would be helpful to include more diverse 3D scenes from different sources to further validate the effectiveness of the proposed method.
2. It is suggested that the authors provide some real-world applications to demonstrate the importance of question clarification in embodied AI.

---

> ### Author Response · Authors · 2025-11-25
> **Rebuttal**
>
> We thank the reviewer for appreciating the merits of our approach and thoughtful comments.  We would like to address their concerns regarding the size of the dataset, error propagation, and baselines.
>
> *Size of the dataset*
>
> We realise that the 263 test instances we prepared might not constitute a very big dataset, but we believe that given the reported effect sizes this test set is sufficient for a meaningful result, showing the effectiveness of our clarification method and the cost reduction compared to the E2E variant.  However, we have now carried out additional experiments, doubling the number of instances by interchanging target and distractor objects within the same scenes.  In these experiments, we used a smaller, open-source foundation model (Qwen3-VL-30B-A3B-Instruct), thereby also addressing a comment from another reviewer.
>
> In the table below, we report Precision, Recall, and F-score before and after clarification, as well as the absolute gain in F-score (Fg).  The results are very similar between the extended and original datasets.  Furthermore, the performance levels when using the Qwen model are lower across the different settings, as expected, but also that the proposed IGSG based clarification method is more effective than in the E2E variant, as before.
>
> | Dataset        |  Condition                  | P-before | R-before | F-before | P-after | R-after | F-after | F-gain |
> | ---------------- | ---------------------------- | -------- | ------- | -------- | -------- | -------- | -------- | -------- |
> | Original       | ambTp-E2E-Qwen      | 39.1   | 87.1   | 53.9    |  66.5  |  69.6  |  68.0   |  +14.1 |
> | Extended    | ambTp-E2E-Qwen      | 39.2   | 87.1   | 54.0    |  65.8  |  68.6  |  67.2   |  +13.2 |
> | Original       | ambTp-IGSG-Qwen    | 40.8   | 98.9   | 57.7    |  86.5  |  80.6  |  83.5   |  +25.8 |
> | Extended    | ambTp-IGSG-Qwen    | 40.4   | 98.9   | 57.3    |  86.8  |  81.4  |  84.0   |  +26.7 |
>
> *Error propagation*
>
> To clarify what happens if “the predicted object ID list is incorrect at the beginning of the dialogue”, if the agent predicts a single object, it considers the user description to be unambiguous and ends the dialogue.  If that prediction is incorrect, the agent has failed to correctly identify the target.  If on the other hand the agent predicts a list of multiple candidates or an empty list, clarification can help to identify the target object, even if the target object is not in the initially predicted list.  It is possible to have the agent confirm a singleton prediction, giving the user the opportunity to correct them if that prediction is wrong, but we leave that for a future version.
>
> *Baselines*
>
> For our specific task of interactive object grounding, there are no other strong baselines that can be included in our evaluation in a straightforward manner.  A variant based on textual scene graphs such as ConceptGraphs  [1] instead of the IGSG could be realised but will most likely perform poorly on the ScanNet based data without the ability to dynamically turn to the images during the reference resolution or captioning subtasks of interactive object grounding.  We therefore only developed the IGSG and E2E system variants and evaluated them.
>
> *References*
>
> [1] Qiao Gu, Ali Kuwajerwala, Sacha Morin, Krishna Murthy Jatavallabhula, Bipasha Sen, Aditya Agarwal, Corban Rivera, William Paul, Kirsty Ellis, Rama Chellappa, et al. Conceptgraphs: Open-vocabulary 3d scene graphs for perception and planning. In 2024 IEEE International Conference on Robotics and Automation (ICRA), pp. 5021–5028. IEEE, 2024

---

> ### Author Response · Authors · 2025-11-25
> **Rebuttal (pt 2)**
>
> Due to space limitations, we address the reviewer's questions in this separate message.
>
> *Diversity of data*
>
> We believe that our current dataset already has a lot of diversity in terms of complexity of the scenes and number of distractors for our task, sufficient for our comparative evaluation.  To provide further insight into how the complexity of the task in terms of number of distractors might affect the results, we have now included results for the cases with one distractor only. As expected, when only considering the cases with one distractor, the scores before clarification are higher and the gains through clarification are smaller, especially in the E2E setting (+9.2 vs +13.2).  More importantly, gains made with the IGSG system are much bigger than with the E2E system, regardless of whether only the 1 distractor cases are considered or not.
>
> | Dataset        |  Condition                  | P-before | R-before | F-before  | P-after  | R-after  | F-after  | F-gain |
> | ---------------- | ---------------------------- | -------- | ------- | -------- | ------- | --------- | -------- | -------- |
> | All                |  ambTp-E2E-Qwen    |   39.2  | 87.1   | 54.0   |  65.8   | 68.6   | 67.2   |  +13.2  |
> | 1 distractor  |  ambTp-E2E-Qwen    |  45.2   | 90.9   | 60.4    |  68.4   | 71.0   | 69.6   |  +9.2    |
> | All                |  ambTp-IGSG-Qwen  |  40.4   | 98.9   | 57.3    |  86.8   | 81.4   | 84.0   |  +26.7  |
> | 1 distractor  |  ambTp-IGSG-Qwen  |  47.0   | 99.5   | 63.9    |  90.9   | 87.9   | 89.3   |  +25.4   |
>
> *Real world applications:*
>
> The ability to use clarification dialogue in object grounding in 3D environments is important in applications where a human operator gives instructions to a mobile robot, for example in a warehouse, factory, or indoor household setting.  To map those instructions to executable robot actions (e.g. “pick up the box”), references to objects (“the box”) need to be grounded in the perceived physical environment, which requires clarification if that reference is ambiguous.

---

> ### Comment · Reviewer_ysk7 · 2025-11-26
>
> Most of my concerns are not addressed appropriately. The authors should prove their statements with experiments instead of **assumptions** (Baselines, Real world applications). Also, can you further clarify what will happen if the agent predicts a list of candidates and all of them are incorrect (I didn't catch your point in the rebuttal)? I will decrease my rating to 2 temporarily. I hope we can have further discussion to improve this work.

---

> > ### Author Response · Authors · 2025-12-03
> >
> > We are not entirely sure which statements and assumptions the reviewer is referring to.
> >
> > Regarding the baselines, we are not disputing that including one that employs a text-only scene graph in some form would be useful.  We merely explained our thinking in prioritising the E2E baseline comparison over this option, given the challenges in implementing such a text-only scene graph based system variant.
> >
> > Regarding the error propagation, from our rebuttal: “If on the other hand the agent predicts a list of multiple candidates or an empty list, clarification can help to identify the target object, **even if the target object is not in the initially predicted list**.”  In other words, if the agent predicts a list of (at least two) candidates that are all incorrect, the dialogue continues and the agent can recover the correct object through further clarification.

---

### Note · Authors · 2026-01-27

I have read and agree with the venue's withdrawal policy on behalf of myself and my co-authors.

---

### Meta-Review · Area_Chair_Kera · 2026-01-06

**Summary:**

This paper presents a benchmark for interactive object grounding. The idea is to find the correct object from many ambiguous ones with the help of language prompting.

The reviewers were concerned about:
* The dataset is too small (200+ samples). The average number of distractors is only 1.33.
* Only a limited type of methods have been evaluated, mostly E2E methods.
* There are limited discussions with other visual grounding data and methods.

**Reviewer Concerns:**

The authors did respond to the above weaknesses, but they could not collect a larger dataset (including raising the difficulty of data) and provide a thorough analysis of other possible methods, either grounding methods or LLM-based methods.

**Reviewer Scores:**

I think the reviewers, after discussion, will arrive at a consensus of weak rejection (4).

---

### Decision · Program_Chairs · 2026-01-26

Reject